# Fistula-Related Cancer in Crohn’s Disease: A Systematic Review

**DOI:** 10.3390/cancers13061445

**Published:** 2021-03-22

**Authors:** Andromachi Kotsafti, Melania Scarpa, Imerio Angriman, Ignazio Castagliuolo, Antonino Caruso

**Affiliations:** 1Laboratory of Advanced Translational Research, Veneto Institute of Oncology IOV-IRCCS, 35128 Padua, Italy; melania.scarpa@iov.veneto.it; 2First Surgical Clinic Section, Department of Surgical, Oncological and Gastroenterological Sciences, University of Padua, 35128 Padua, Italy; imerio.angriman@unipd.it; 3Department of Molecular Medicine DMM, University of Padua, 35121 Padua, Italy; ignazio.castagliuolo@unipd.it; 4Gastroenterology Unit, ULSS2 Marca Trevigiana, Montebelluna Hospital, 31044 Montebelluna, Italy; antonino.caruso@aulss2.veneto.it

**Keywords:** squamous cell carcinoma, perianal fistulas, anal fistula-related cancers, Crohn’s disease

## Abstract

**Simple Summary:**

Cancer arising at the site of a chronic perianal fistula is rare in patients with Crohn’s disease. The relationship between perianal fistula in CD (Chron’s disease) and SCC (squamous cell carcinoma) development is not clear but chronic inflammation of ano-rectal mucosa, delayed wound healing and cell turnover may play important roles. The aim of this systematic review was to determine the clinical characteristics of patients with squamous cell carcinoma arising from perianal fistula in CD, the surgery and oncological treatment, the role of HPV infection, immunosuppression and the survival of these patients. Fistula-related carcinoma in CD can be very difficult to diagnose. An early diagnosis has the potential to improve the outcome of disease.

**Abstract:**

Perianal fistulizing Crohn’s disease is a very disabling condition with poor quality of life. Patients with perianal fistulizing Crohn’s disease are also at risk of perianal fistula-related squamous cell carcinoma (SCC). Cancer arising at the site of a chronic perianal fistula is rare in patients with Crohn’s disease and there is a paucity of data regarding its incidence, diagnosis and management. A systematic review of the literature was undertaken using Medline, Embase, Pubmed, Cochrane and Web of Science. Several small series have described sporadic cases with perianal cancer in Crohn’s disease. The incidence rate of SCC related to perianal fistula was very low (<1%). Prognosis was poor. Colorectal disease, chronic perianal disease and HPV infection were possible risk factors. Fistula-related carcinoma in CD (Chron’s disease) can be very difficult to diagnose. Examination may be limited by pain, strictures and induration of the perianal tissues. HPV is an important risk factor with a particular carcinogenesis mechanism. MRI can help clinicians in diagnosis. Examination under anesthesia is highly recommended when findings, a change in symptoms, or simply long-standing disease in the perineum are present. Future studies are needed to understand the role of HPV vaccination in preventing fistula-related cancer.

## 1. Introduction

Perianal disease is a very disabling condition and a poor prognostic indicator in patients with Crohn’s disease (CD). About 25% of patients develop perianal fistulas during their life [1]. A long disease duration increases the cumulative incidence of perianal fistulas: 12% after 1 year, 15% after 5 years, 21% after 10 years and 26% after 20 years [2,3,4]. Perianal lesions can precede the diagnosis of CD and anal stricture is a risk factor for fistulizing disease [4].

Perianal fistulas are more frequent in patients with colonic CD with rectal involvement (92%) [3]. Patients with perianal Crohn’s disease have a 0.7% incidence of carcinoma [5]. The development of neoplasia in chronic CD fistulas is considered rare and the risk is related to perianal disease duration [4]. The exact etiology is not fully known and the literature is low on data concerning the incidence, diagnosis and treatment of this condition. Most reports consist of single cases or small series.

Two possible histological types are more frequent: squamous cell carcinoma (SCC) and adenocarcinoma. The research in this work focuses on the former. The relationship between perianal fistula in CD and SCC development is not clear but chronic inflammation of ano-rectal mucosa, delayed wound healing and cell turnover may play important roles [6,7]. The role of immunosuppression is conflicting. The role of human papilloma virus (HPV) as a risk factor has recently increased. Fistula-related cancer is not associated with specific signs or symptoms; the malignant lesion could be in the inaccessible fistula tract and a surveillance protocol does not currently exist. For these reasons, the prognosis of these patients is generally poor and reflects a diagnostic delay [6,7,8].

The objective of this review is to summarize the clinical characteristics of patients with squamous cell carcinoma arising from perianal fistula in CD, the surgery and oncological treatment, the role of HPV infection, immunosuppression and the survival of these patients.

## 2. Materials and Methods

A systematic review of the literature was performed for case series and reports. Only data from original articles were extracted, referring to the following parameters: demographic data, risk factors, outcome, clinical manifestations, radiological assessment (computed tomography (CT) or magnetic resonance (MR)), treatment (chemotherapy vs. surgery), method of diagnosis and the HPV status of inflammatory bowel disease (IBD) patients presenting SCC related to chronic perianal fistula. The methodology undertaken was based on the PRISMA statement guidelines [9].

### 2.1. Search Strategy

A medical literature search was conducted in September 2020 using Embase, Medline, Web of Science, the Cochrane Database of Systematic Reviews and the Cochrane Central Register of Controlled Trials (CENTRAL) (from December 1991 to October 2020). The following Medical Subject Headings (MeSH) were used: Crohn’s disease, perianal fistula and squamous cell carcinoma. Only clinical studies in the English and Italian languages were considered. Inclusion criteria for the analysis were: CD patients with perianal fistulizing disease and SCC arising from fistula. Patients with anal cancers, rectal cancers and other types of fistula in CD were excluded. A manual cross-reference search for eligible papers was also performed to identify additional relevant articles and studies which satisfied our criteria. Unpublished data and abstracts were not considered. The initial search identified 75 articles, 57 after removing duplicates. Forty of them were excluded due to the following reasons: 30—unrelated to our topic; 1—not relevant; 1—missing data; 2—SCC did not arise from the fistula; 1—data overlapped with that of other articles; 1—technical review of MR features of fistulas and perianal cancer; and 4—findings published only in abstracts (Figure 1). A total of 36 cases from 17 studies were analyzed by the end of our review.

### 2.2. Data Extraction

Two investigators (A.C. and A.K.) independently extracted the following data: patient characteristics, risk factors, clinical manifestations, type of surgical approach, treatment, method of diagnosis, outcome and HPV status of IBD patients presenting SCC related to chronic perianal fistula. The two authors reviewed all eligible studies independently until full concordance was achieved for all variables assessed.

### 2.3. Statistical Analysis

We could only perform a descriptive analysis given the small size of the sample and the study’s retrospective nature. Continuous variables were expressed as mean or median (range).

## 3. Results

Details of our literature selection are shown in Figure 1. Seventeen studies met the inclusion criteria and were included in this review [7,10,11,12,13,14,15,16,17,18,19,20,21,22,23,24,25]. The characteristics of included studies are presented in Table 1, Table 2 and Table 3. A total of 36 patients with a squamous cell carcinoma arising from perianal fistula in CD were evaluated. A total of 21 were female (58.3%) and 15 were male (48.7%). The median age was 48 years (range 28 to 76 years). The women were younger (mean 46.2 years, median 45 years; range 30–76) than the men (mean 50 years, median 52 years; range 28–66). The mean of years from Crohn’s disease to cancer diagnosis was 20.81 years (median 20, range 3 to 48 years). The median duration of disease to cancer was 20 years for females (range 3 to 48 years) and 20 years for males (range 3 to 33 years). The average of years from fistula to SCC diagnosis was 10.85 (median 10, range 3 to 33 years). Data on smoking was lacking because this factor was only specified in 6 cases (4 patients were active and 2 were previous smokers). The analysis was unfeasible due to lack of data. All patients had perianal disease. The extension of disease was ileocolic in 15 patients (51.7%), 9 patients had a colonic localization of disease (31%), 3 only had perianal disease (10.3%), 2 anorectal disease (7%) and data were not indicated for 7 patients. The c/pT stage was only available for 8 patients: stage I (*n* = 1), stage II (*n* = 3), stage III (*n* = 2) and stage IV (*n* = 2). In this study population, data on medical therapy for CD were available in 18 patients. A total of 6 patients received steroids, 5 were treated with immunomodulators (1 methotrexate, 1 6-mercaptopurine and 3 azathioprine) and 7 were in biological therapy with anti-TNFalpha. A total of 18 (50%) patients underwent prior surgery (12 for CD and 6 for SCC but before oncological treatment) and 13 (37.14%) after chemoradiation. One patient’s surgical history was not available (for details see Table 2). Among all patients with SCC, 23 received oncological treatment with Nigro protocol which included radiation (50–55 Gy) and chemotherapy with 5-fluorouracil and mitomycin. A total of 5 patients only received radiotherapy and 7 no therapy. Treatment was not indicated for 1 patient. A total of 30 (93.75%) patients referred symptoms related to their perianal fistula, 2 (6.25%) patients presented asymptomatic chronic fistula, while data were unknown in 4 patients. The most frequent symptom that led to diagnosis of SCC was a severe or increasing perianal pain (17 patients, 56.66%). A total of 2 (6.66%) patients presented sepsis, 2 perianal abscess (6.66%), 2 pus discharge from fistula (6.66%), 2 rectal bleeding (6.66%), 1 patient referred diarrhea (3.33%), 1 patient presented discharge of stool from fistula (3.33%), 2 presented anal stricture (6.66%) and 2 perianal mass (3.33 %). HPV status was only present in 12 cases. A total of 8 patients were positive while the other 4 were negative. The diagnosis of SCC was performed with a biopsy taken from the fistula tract in 30 patients (88%), in 4 cases (12%) the tumor was discovered during surgery, while data were unknown for 2 patients. Thirteen patients underwent CT or MR to assess the local or metastatic spread of disease. The outcome was specified in 22 patients: 11 died within 2 years from diagnosis of SCC and 11 survived. Long-term follow-up (5 years) was available in 2 patients, while the follow-up period for the others patients was of no longer than 3 years.

## 4. Discussion

Crohn’s disease is a chronic inflammatory disease that can involve the entire gastrointestinal tract. The fistulizing phenotype can affect the perianal region from 5% to 40% of cases and it is more common in patients with severe rectal and colon involvement. Colonic and rectal disease are the most important risk factors for the development of perianal fistulas in patients [26,27,28]. Early onset of disease, long-standing disease (> 10 years), severe chronic colitis, chronic fistula and stenosis are important risk factors for carcinogenesis of a fistula tract [8].

The etiology of fistulas may derive from inflamed or infected anal glands and/or penetration of fissures or ulcers in the rectum or anal canal of CD patients. The most common symptoms of perianal fistulas are pain, purulent discharge and fecal incontinence with poor quality of life for patients [26,27,28]. The diagnosis of neoplasia in chronic perianal fistulas is difficult and is often delayed because symptoms are non-specific, therefore, the biopsy is usually only performed at a late stage of disease [29]. Chronic inflammation of the rectum in CD patients with perianal disease can increase the risk of cancer, although cancer arising from a fistula is a rare condition [6]. A meta-analysis of 20 clinical studies including more than 40,000 CD patients showed that CD is a risk factor for intestinal cancer but the incidence of perianal cancer arising from a fistula was very low at 0.2/1000 patients years (95% CI, 0.0/1000–0.4/1000) [30]. The risk of fistula-associated cancer is related to disease duration. In a Dutch study involving more than 6000 CD patients, cancer arising from a fistula was observed in only four patients and the histological type was adenocarcinoma. Malignancies developed 25 years after CD diagnosis, and 10 years after fistula diagnosis [29]. There is limited literature on squamous cell cancer (SCC) arising from a fistula in CD.

A meta-analysis of Thomas et al. showed 61 cases of neoplasia arising in a fistula in CD. In this population, 31% of cancers were SCC, while the rest were adenocarcinomas (59%). Sixty-one percent of patients were female and were younger than the males [31]. Benjelloun and colleagues analyzed the clinical data of 21 CD patients with perianal disease and SCC. Thirteen were female with a mean age of 45.1 (range, 28–76) years [14].

Our research identified 17 studies with a total of 36 patients. Women were predominant (58.3%) and younger than men (46 vs. 50 years), in agreement with the data in the literature. Another important consideration is disease duration. Most patients have a long history of disease with chronic perianal fistula and poor quality of life. In our population, the mean duration of CD is 21 years with a time lapse of over 10 years from fistula to cancer diagnosis. As suggested by several authors, chronic inflammation may be the “primum movens” of malignant transformation in the sequence dysplasia-carcinoma [6,8,21]. In the general population, HPV infection, in particular types 16 and 18, is closely associated with perianal and anal cancer. The prevalence of anal HPV infection in the IBD population is high, particularly in patients with perianal involvement. A hypothesis of pathogenetic mechanism is that lesions of anal mucosa and epithelialization of fistula may allow direct access of HPV to keratinocytes, promoting HPV carcinogens [17,32,33]. In a study conducted on 26 IBD patients, 81% showed HPV anal infection and 43% of patients with dysplasia were treated with immunosuppressive therapies [34]. A report on 18 IBD patients showed that anal SCC was associated with HPV infection, perianal disease and long-standing disease. Fifty percent of patients (3/6) with SCC and HPV infection had perianal disease [35]. Kuhlgatz et al. initially reported the presence of HPV 6 and, subsequently, HPV 16 in a CD patient with SCC arising from a perianal fistula [23]. In 2017, Lightener et al. performed a retrospective study on 7 perianal CD patients and SCC, finding a high prevalence (42%) of HPV infection [18]. In our population, we found 8 patients with HPV infection but the data were lacking in other cases. The number would probably have been higher if the presence of HPV had been evaluated in more cases. These data suggest a strong association between HPV infection and SCC in fistulizing CD. The real role of immunosuppression is debated in the literature. The medical management of fistulizing disease commonly promotes the use of immunosuppressive therapy, in particular anti-TNF alpha. Several reports in the literature showed a major incidence of SCC in immunosuppressed patients. Ball et al. assumed that chronic immunosuppression may be a possible mechanism of carcinogenesis in perianal fistula [36]. In a population-based case-control study of NMSC (non-melanoma skin cancer), the risk of SCC increased twofold with the use of oral glucocorticoids for a short period of less than six months [37]. Recent studies showed that thiopurines used alone or in combination with anti-TNFs increase the risk of NMSC, particularly of SCC [8,38]. In our population, 14 patients received immunosuppression (steroids, immunomodulators or biologics); however, we cannot speculate on a clear relationship with SCC. All these patients had long-standing disease with chronic perianal fistula requiring immunosuppressive treatment. In our opinion, these data reflect the immunological imbalance of IBD patients but, in reality, we are far from confirming that immunosuppression can induce carcinogenesis in the fistula tract, also because another consideration is that fistulizing disease is considered an independent risk factor for neoplasia in IBD [38]. The outcome of patients with SCC arising from a fistula was poor, with a mean survival rate of 5 years after diagnosis [14]—a fact which is also corroborated in our patients, in line with literature reports. Anal stricture and pain limiting examination without anesthesia, localization of lesions inside inaccessible fistula tracts and non-specific signs and/or symptoms complicate and often delay diagnosis, thus worsening the prognosis [6,8,31]. MR is reported to be useful not only for fistulizing disease but also for the diagnosis of perianal carcinoma. Lad et al. showed that the combination of an irregular inner wall and delayed enhancement were typical of patients with perianal neoplasia. A thickening of the fistula wall was also present, but this alone was not enough to differentiate between cancer and inflammation. Only two patients had an SCC. In both cases, the radiological picture showed a double-layered enhancement pattern [39]. We found that for patients where a biopsy was performed for diagnosis, it is sometimes specified that this procedure was performed during exploration under anesthesia (EUA) or during surgery. Thirteen patients underwent an imaging technique to support the diagnosis of cancer or to stage the disease. In our opinion, in the case where a patient has a change in symptoms or long-standing perianal disease, it could be useful to perform a tissue biopsy if the MR or CT scan is inconclusive.

The treatment of fistula-related SCC includes chemoradiation and/or surgery. In our series, 23 patients underwent Nigro protocol involving chemotherapy with mitomicyn plus 5-fluorouracil and radiation (50–55 Gy). Eight patients underwent surgery (APR) for residual disease at the end of therapy. Six patients underwent surgery before chemoradiation. This strategy may be useful to treat an active perianal disease in patients with a poor tolerance for chemoradiation and in order to prevent local radiotherapy complications. No treatment was undertaken when the disease was in an advanced state.

Our work has several limitations. The quality of the review is influenced by the study’s retrospective design and the small sample size. We only found case reports on SCC arising from fistula in CD. We only identified 17 studies with a small number of patients (from one to nine) for a total of 36 cases. Data on these reports even lacked clinical information and thus we could only perform a descriptive analysis. Our study’s evaluation should take these limits into consideration since these limitations have probably influenced final quality and accuracy. In our inclusion criteria, we considered the most important elements that influence the natural history of disease, particularly CD duration, modality of diagnosis, treatment, outcome and the HPV status of patients. This work’s strength lies in the fact that we analyzed a rare condition in order to emphasize the need to establish a protocol for these patients. We underline the increasing role of HPV infection and the major risk for female patients with fistulizing disease. In this clinical scenario, it became very important to test patients for HPV before starting biological therapy for fistulizing disease. We also summarized surgical treatments and found that a plausible strategy could be early surgery, when possible, in order to avoid radiotherapy complications in patients with active perianal CD.

## 5. Conclusions

SCC arising from fistula in CD patients is a rare condition. We underlined the importance of early diagnosis to improve the outcome of disease. Symptoms are non-specific and, for this reason, we urge clinicians to perform imaging (in particular, MR with contrast enhancement) in patients with long-standing fistulizing CD. Furthermore, an exploration and biopsy under anesthesia is mandatory if the patient’s condition worsens or new symptoms manifest. Early diagnosis could allow early chemoradiation treatment with a better outcome. The role of HPV infection is increasing, particularly with the use of biological therapy. For this reason, it is recommended that female patients with fistulizing CD receive routine pap smears and, where possible, a HPV vaccination.

## Figures and Tables

**Figure 1 cancers-13-01445-f001:**
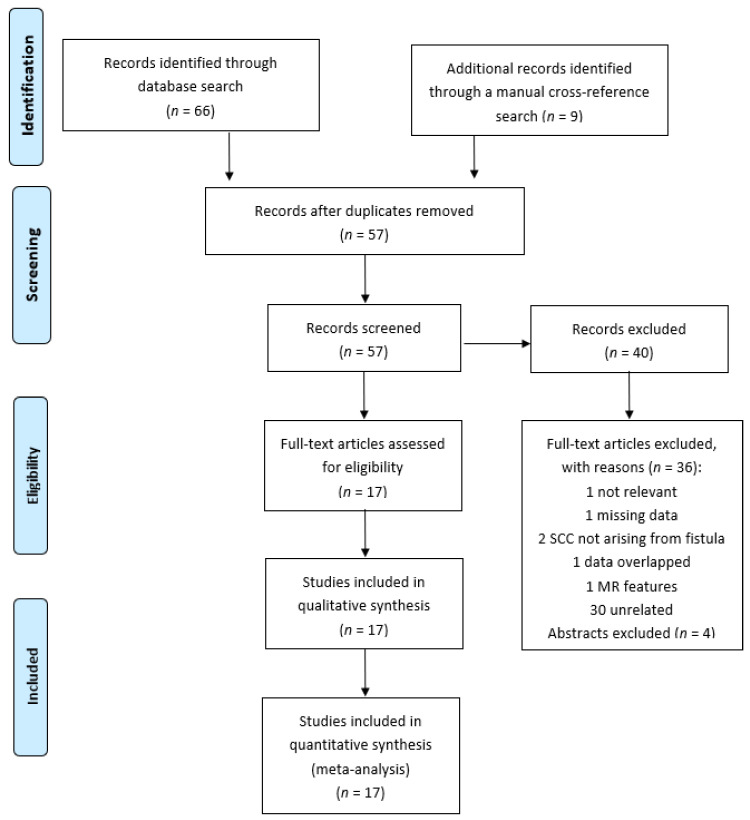
Prisma flowchart of study selection.

**Table 1 cancers-13-01445-t001:** Demographic characteristics of CD patients with SCC arising from perianal fistula.

Ref. Number	Institution (Inclusion Period)	Patient No.	Age	Sex	Smoking Status	Duration of CD (Years)	Presence of Perianal Disease	Extent of CD
[22]	Mount Sinai Medical Center, New York, NY, USA (1976–1981)	1	36	F	NA	3	Y	colonic
		2	40	F	NA	14	Y	ileocolic
[12]	Departments of Therapeutics and Surgery, City Hospital, Nottingham, UK (1982)	3	30	M	NA	4	Y	ileocolic, perianal

[21]	Cleveland Clinic, Cleveland, OH, USA (1972–1983)	4	59	M	NA	30	Y	rectal
		5	65	F	NA	21	Y	ileocolic
[13]	St Mark’s Hospital, London, UK (1940–1991)	6	38	F	NA	18	Y	anorectal

[15]	George Washington University, Washington, DC, USA (1991)	7	49	F	NA	23	Y	perianal

[20]	Lenox Hill Hospital, New York, NY, USA (1983–1997)	8	47	M	NA	>10	Y	NA
		9	38	M	NA	>20	Y	NA
		10	30	F	NA	8	Y	NA
		11	31	F	NA	20	Y	NA
[11]	Buffalo, New York, NY, USA	12	59	M	NA	33	Y	ileocolic
[25]	Mount Sinai Medical Center, New York, NY, USA	13	36	F	NA	12	Y	colonic
[24]	Saint Louis University Health Sciences Center, St. Louis, MO, USA	14	60	M	NA	19	Y	colonic
[7]	Yale University School of Medicine, New Haven, CT, USA	15	76	F	NA	>30	Y	colonic
[23]	Albert Schweitzer Hospital, Northeim, Germany	16	53	M	NA	27	Y	colonic
[17]	Mount Sinai Hospital, Toronto, ON, Canada	17	31	F	Yes	18	Y	NA
		18	50	F	No	30	Y	NA
		19	46	F	Yes	26	Y	NA
[16]	Yonsei University, Seoul, South Korea (2008)	20	43	F	NA	11	Y	ileocolic, perianal
[14]	University hospital Hassan II. Fez., Morocco (2012)	21	47	M	NA	20	Y	perianal
[10]	Mount Sinai Medical Center, New York, NY, USA (2003–2015)	22	64	M	NA	13	Y	ileocolic
		23	50	M	NA	33	Y	ileocolic
		24	45	F	NA	>10	Y	ileocolic
		25	46	M	NA	25	Y	ileocolic
		26	65	F	NA	31	Y	ileocolic
		27	31	F	NA	16	Y	ileocolic
		28	53	M	NA	>10	Y	perianal
		29	33	F	NA	12	Y	ileocolic
		30	70	F	NA	48	Y	ileocolic
[18]	Mayo Clinic, Rochester, NY, USA (1995–2016)	31	51	F	Active	30	Y	ileocolic
		32	66	M	Prior	20	Y	colonic
		33	52	F	Never	22	Y	colonic
		34	28	M	Prior	14	Y	colonic
		35	54	F	Never	35	Y	ileocolic
[19]	University Hospital, LMU Munich, Munich, Germany	36	54	M	Active	20	Y	colonic

NA: not available; Y: Yes.

**Table 2 cancers-13-01445-t002:** Demographic and clinical characteristics of CD patients with SCC arising from perianal fistula.

Patient No.	Biologics, Steroids, Immunomodulators	Past Surgery	Years from CD to Cancer Diagnosis	Years from Fistula to Cancer Diagnosis	HPV Status	Imaging
1	NA	NO	3	NA	NA	NA
2	NA	NO	14	NA	NA	NA
3	NO	NO	4	0	NA	CT
4	NA	NO	30	NA	NO	NA
5	NA	NO	23	NA	NO	NA
6	aza	NO	15	15	NA	NA
7	steroids, aza	APR	23	16	NA	CT
8	NA	NO	10	3	NA	NA
9	NA	fistulotomy, ileocolic resection	3	10	NA	NA
10	NA	NO	10	11	NA	NA
11	NA	Hartmann	20	5	NA	NA
12	NO	Ileo-cecal resection, colostomy	33	33	NA	CT
13	steroids	proctectomy	12	3	NA	CT
14	steroids	diverting colostomy	19	Unknown	NA	CT
15	NA	diverting colostomy	>30	Unknown	NA	MR
16	steroids, aza	subtotal colectomy	27	27	HPV 6 DNA HPV 16 DNA	CT
17	NA	total proctocolectomy, vaginectomy, flap	18	18	Yes	MR, CT
18	NA	APR	30	13	No	MR, CT
19	NA	NO	26	NA	Yes	MR, CT
20	steroids, IFX	ileal resection, right hemicolectomy, ileostomy	11	10	NA	MR, CT
21	No	diverting colostomy	20	20	NA	MR
22	aTNF	NO	13	4	NA	NA
23	aTNF	NO	33	7	NA	NA
24	NO	NO	>10	0	NA	NA
25	aTNF	NO	25	NA	NA	NA
26	NO	NO	31	5	NA	NA
27	aTNF	NO	16	3	NA	NA
28	NO	APR	>10	NA	NA	NA
29	NO	TPC	12	NA	NA	NA
30	aTNF	NO	48	14	NA	NA
31	prednisone certolizumab	several small bowel resections	30	NA	Yes	NA
32	NO	left colectomy and partial proctectomy; APR/VRAM for CD with incidental finding of ASCC	20	NA	Yes	NA
33	6-MP	NO	22	NA	Yes	NA
34	MTX	EUA/seton	14	NA	Yes	NA
35	NO	proctocolectomy with ileostomy (1970’s)	35	NA	Yes	NA
36	NA	NO	20	NA	NO	CT

aza: azathioprine; MTX: methotrexate; 6-MP: 6-mercaptopurine; IFX: infliximab; aTNF: anti-TNF; APR: abdominal perineal resection; TPC: total proctocolectomy; VRAM: right vertical rectus abdominis myocutaneous flap; EUA: exam under anesthesia; ASCC: anal squamous cell carcinoma; NA: not available, HPV: Human Papilloma Virus; MR: Magnetic Resonance; CT: Computed Tomography.

**Table 3 cancers-13-01445-t003:** Clinical characteristics of CD patients with SCC derived from perianal fistula.

Patient No.	Stage	RT	CT	Residual Disease 6 Months	Surgery after CT-RT	Treatment of Recurrent Disease	Symptoms	Cancer Treatment	Outcome	Diagnosis
1	NA	Yes	No	No	colectomy and APR	No	pain	RT+colectomy+APR	Alive 2 y	NA
2	NA	No	No	No	No	RT	abscess	Subtotal Colectomy (1968) and APR 1982	Alive 1.5 y	NA
3	NA	No	No	NA	No	No	perianal mass	Right hemicolectomy, excision of anal lesion	Alive	biopsy
4	NA	Yes	No	Yes	proctectomy	No	severe pain	RT+proctectomy	Died 10 mo	biopsy
5	NA	Yes	No	No	No	No	diarrhea	RT	Died 2 y	EUA
6	NA	Yes	Yes	No	No	No	anal lump, anal stricture	CT+RT	Alive 6 mo	EUA
7	NA	Yes	No	NA	No	No	bleeding, perianal mass	RT	lost to FU	biopsy
8	NA	Yes	Yes	NA	No	No	severe pain	CT+RT	Died 6 mo	EUA
9	NA	Yes	Yes	NA	excision	local excision	persistent fistula	CT+RT+excision	Alive	biopsy
10	NA	Yes	Yes	NA	APR	No	severe pain	CT+RT+APR	Alive	biopsy
11	NA	Yes	Yes	NA	APR	No	severe pain	CT+RT+APR	Died 1.5 y	biopsy
12	NA	No	No	No	No	local excision	perianal mass	debridement, proctectomy	Died 9 mo	surgery
13	NA	Yes	Yes	No	ileostomy	RT	NA	CT+RT+Proctectomy	Alive 6 y	surgery
14	NA	No	No	NA	NA	NA	pus discharge	Debridement	lost to FU	biopsy
15	NA	No	No	No	No	No	septic shock	No treatment	Died	biopsy
16	II	Yes	Yes	No	APR	local excision	pus discharge	APR	NA	biopsy
17	IV	Yes	Yes	NA	No	NA	pain	CT+RT+TPC	Alive 26 mo	surgery
18	II	Yes	Yes	NA	No	NA	pain	CT+RT+APR	Died 23 mo	surgery
19	II	Yes	No	No	APR, vaginectomy, flap	NA	pain	RT+APR	Alive 37 mo	biopsy
20	NA	Yes	Yes	Yes	No	No	severe anal stricture, pain	CT+RT	NA	biopsy
21	NA	No	No	No	No	No	perianal abscess	none	Died 3 mo	biopsy
22	NA	Yes	Yes	NA	No	No	asymptomatic chronic fistula	CT+RT	NA	biopsy
23	NA	Yes	Yes	NA	No	No	increasing pain and drainage	CT+RT	NA	biopsy
24	NA	Yes	Yes	NA	APR	No	increasing pain and drainage	CT+RT+APR	NA	biopsy
25	NA	Yes	Yes	NA	APR	No	unknown	CT+RT+APR	NA	biopsy
26	NA	Yes	Yes	NA	No	No	increasing pain and ulceration	CT+RT	NA	biopsy
27	NA	Yes	Yes	NA	No	No	asymptomatic chronic fistula	CT+RT	NA	biopsy
28	NA	Yes	Yes	NA	No	No	unknown	APR+CT+RT	NA	biopsy
29	NA	Yes	Yes	NA	No	No	unknown	TPC+RT+CT	NA	biopsy
30	NA	No	No	NA	No	No	increasing pain, severe sepsis	No treatment	Died 1 y	biopsy
31	IIIB	Yes	Yes	No	APR/VRAM (for CD, no recurrence)	NA	perianal pain	RT+CT+APR	Alive 5 y	biopsy
32	IIIA	Yes	Yes	NA	No	CT-RT	bleeding	CT+RT	Died 6 mo	biopsy
33	I	Yes	Yes	No	No	NA	perianal pain	CT+RT	Alive 5 y	biopsy
34	NA	Yes	Yes	Yes	APR	NA	perianal pain	CT+RT+APR	NA	biopsy
35	IV	Yes	Yes	Yes	APR/VRAM IORT	Chemotherapy	perianal pain	CT+RT+APR+IORT	Died 6 mo	biopsy
36	NA	NA	NA	NA	NA	NA	discharge of stool through perianal fistulas, weight loss	APR, excision of perineal tumor	NA	biopsy

CT-RT: chemoradiation; RT: radiotherapy; CT: chemotherapy; APR: abdominal perineal resection; VRAM: right vertical rectus abdominis myocutaneous flap; IORT: intraoperative radiotherapy; mo: months; FU: follow-up.

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
