# Peer review of "Fistula-Related Cancer in Crohn’s Disease: A Systematic Review"

_cancers, 2021, doi:10.3390/cancers13061445_

Round 1

Reviewer 1 Report

The subject is very interesting because cancers arising in a site of perianal fistula due to Crohn's disease are rare and there is a paucity of data regarding this issues.

The review is well done and the manuscript is well written. 

However, Prisma flow-chart should be corrected: record screened (57), excluded (40), assessed for eligibility - 22?

After this correction my recomendation is: acceptation

Author Response

We thank the reviewer for stating that the subject is interesting and that it is well written. We apologize for the mistake in PRISMA flow-chart and as suggested, we corrected it (please see the attachment).

Reviewer 2 Report

The aim of this review was to summarize the clinical characteristics of patients with squamous cell carcinoma arising from perianal fistula in CD, the surgery and oncological treatment, the role of HPV infection, immunosuppression, and survival.

The authors underlined the importance of early diagnosis to improve the outcome. MRI is recommended to obtain an early diagnosis as well as biopsy under anesthesia in case of a suspicious lesion. HPV vaccination is also recommended.

A major strength of this study was that the PRISMA guidelines were followed. An earlier review from 2018 did not use PRISMA. (Yamamoto T, Kotze PG, Spinelli A, Panaccione R. Fistula-associated anal carcinoma in Crohn's disease. Expert Rev Gastroenterol Hepatol. 2018 Sep;12(9):917-925. doi: 10.1080/17474124.2018.1500175)

Author Response

We thank the reviewer for the comments and for the positive evaluation on the use of PRISMA guidelines.